



**Tropospheric warming over the North Indian Ocean caused by the South Asian anthropogenic aerosols: possible implications**
Suvarna Fadnavis[1*], Prashant Chavan[1], Akash Joshi[2], Sunil Sonbawne[1], Asutosh Acharya[3],
Panuganti C S. Devara[4], Alexandru Rap[5], Rolf Müller[6]
[1]Indian Institute of Tropical meteorology, MoES, Pune, India
[2]Indian Institute of Technology, Kharagpur, India
[3]Indian Institute of Technology, Bhubneshwar, India
[4]Centre of Excellence in ACOAST/ACESH, Amity University Haryana (AUH), Gurugram
122413, India
[5]School of Earth and Environment, University of Leeds, Leeds, United Kingdom
[6]Forschungszentrum Jülich GmbH, IEK-7, Jülich, Germany
Corresponding author: Suvarna Fadnavis
Corresponding author email: suvarna@tropmet.res.in
**Abstract**
Atmospheric concentrations of South Asian anthropogenic aerosols and their transport play a
key role in the regional hydrological cycle. Here, we use the ECHAM6-HAMMOZ
chemistry-climate model to show the structure and implications of the transport pathways of
these aerosols during spring. Our simulations indicate that large amounts of anthropogenic
aerosols are transported from South Asia to the North Indian Ocean (the Arabian Sea and
North Bay of Bengal). These aerosols are then lifted into the upper troposphere and lower
stratosphere (UTLS) by the convection over the Arabian Sea and Bay of Bengal. In the
UTLS, they are further transported to the southern hemisphere (30-40°S) and downward into
the troposphere by the secondary circulation induced by the aerosol changes. The
carbonaceous aerosols are also transported to the Arctic and Antarctic producing local
heating ($0.002 – 0.05$ K d$^{-1}$).





The presence of anthropogenic aerosols causes negative radiative forcing (RF) at the TOA (-
0.90±0.089 W m$^{-2}$) and surface (-5.87±0.31 W m$^{-2}$) and atmospheric warming (+4.96±0.24 W
m$^{-2}$) over South Asia (60° E - 90° E, 8° N - 23° N), except over the Indo-Gangetic plain
(75°E - 83° E, 23° N - 30° N) where RF at the TOA is positive (+1.27±0.16 W m$^{-2}$) due to
large concentrations of absorbing aerosols. The carbonaceous aerosols produced in-
atmospheric heating along the aerosol column extending from the boundary layer to the
UTLS (0.01 to 0.3 K d$^{-1}$) and in the stratosphere globally (0.002 to 0.012 K d$^{-1}$). The heating
of the troposphere increases water vapor concentrations, which are then transported from
highly convective region (i.e. the Arabian Sea) to the UTLS (increasing water vapor by 0.02 –
0.06 ppmv).
Keywords: South Asian Anthropogenic aerosols; warming over the Arabian Sea; transport of
aerosols and water vapor to the UTLS in spring.



## 1. Introduction


Understanding the variability of anthropogenic aerosol loading over the North Indian
Ocean is of utmost importance since (1) it regulates the Asian hydrological cycle via
modulating atmospheric convection, heating rates, and moisture transport (Ramanathan et al.
2005; Corrigan et al., 2008; Budhavant et al., 2018, Meehl et al., 2008), and (2) it leads to
adverse impacts on marine ecosystems (Mahowald et al., 2018; Collins et al., 2019). Several
observations indicate that the aerosol loading over the North Indian Ocean during the spring
season is strongly influenced by South Asian aerosols. Aircraft measurements during the
Indian Ocean Experiment (INDOEX) (February–March 1999) showed the presence of a thick
layer (surface to 3.2 km) of anthropogenic aerosols (BC~14 %, sulfate 34 %, ammonium 11
%) over the North Indian Ocean (Dickerson et al., 2002; Mayol-Bracero et al., 2002) with
sources over South Asia. Several other in situ observations, e.g. over the Maldives during
November 2014 – March 2015, show that air masses arising from the Indo-Gangetic Plain
contain very high amounts (97 %) of the elemental carbon in the fine mode. Other
anthropogenic species such as organic carbon, non-sea-salt, potassium, and ammonium (70–
95%) were also observed in the fine mode (Bhudhvant et al., 2018). Observations from the
Geosphere-Biosphere Programme over the Bay of Bengal during spring (March 2016) also
show abundant anthropogenic aerosols (sulfate and nitrate) having sources over the Indo-
Gangetic plain (Nair et al., 2017).
The aerosol loading over South Asia has been increasing at an alarming rate (rate of
increase in AOD 0.004 per year during 1988 – 2013) (Babu et al., 2013). For the last two
decades, the AOD increase (by 12 %) over south Asia has been attributed to the strong
increase in anthropogenic aerosols (sulfate, black carbon, and organic carbon), while natural
aerosol remained unchanged (Ramachandran et al., 2020a). The major sources of



anthropogenic aerosols are the combustion of domestic fuels, industrial emissions,
transportation, and open burning (Paliwal et al., 2016). The growth of the economy of India
led to a 41 % increase in BC and 35 % in OC from 2000 to 2010 (Lu et al., 2011). The
emissions of sulfur dioxide ($SO_2$) which leads to the production of sulfate aerosols have
doubled during 2006 – 2017 (Fadnavis et al., 2019). Figure 1 a-c shows the annual mean
emission of BC, OC, and sulfate aerosols over South Asia in 2016, with high emissions over
the Indo-Gangetic Plain (BC  7E-12 – 17E-12 Kg m$^{-2}$ S$^{-1}$, OC: 25E-12 - 70E-12 Kg m$^{-2}$ S$^{-1}$,
sulfate: 2E-12 - 5E-12 Kg m$^{-2}$ S$^{-1}$). Higher amounts of aerosols over the Indo-Gangetic Plain
are associated with densely populated regions and industrial and vehicular emissions
(Karambelas et al., 2018, Fadnavis et al., 2019). Past studies also show substantially higher
amounts of aerosols over North India compared to rest of the Indian region (Ramachandran et
al., 2020b, Fadnavis et al., 2013, 2017a, 2017b). Over the Indo-Gangetic plain, these
emissions show a peak in spring (Fig. 1d), with increases for BC of 0 – 3 %, OC 0 – 8.7 %,
and sulfate 0 – 0.2 %, compared to annual means. This peak in emissions in spring is to a
large extent driven by springtime agricultural crop burning and biomass burning activity
(Chavan et al., 2021).
While the presence of sulfate aerosols lead to a cooling of the atmosphere below due to
their strong scattering properties, carbonaceous aerosols produce atmospheric warming via
absorption of solar radiation (Fadnavis et al., 2019, Penner et al., 1998).  Previous studies
showed that the doubling of carbonaceous aerosols loading over Asia (10° S – 50° N, 65° E–
155° E) led to significant atmospheric warming (in-atmospheric RF 5.11W m$^{-2}$, Fadnavis et al.
2017b).


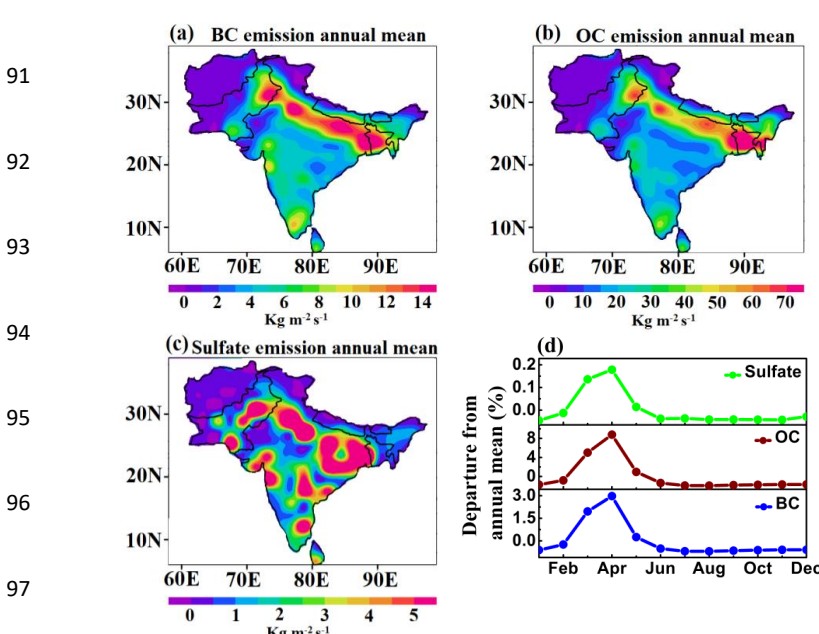

Figure 1: Spatial distribution of year 2016 annual mean total emission (kg m$^{-2}$ S$^{-1}$) of (a) BC, (b) OC, (c) Sulfate aerosols, (d) time series of monthly departure from annual mean total emissions (%) of BC, OC, and Sulfate aerosols averaged over Indo-Gangetic plain (23° – 30° N, 78 – 90° E).

During spring, the prevailing convective instability over the Bay of Bengal and the Arabian Sea transports aerosol from the boundary layer to the upper troposphere (Romatschke and Houze, 2011). Airborne observations during winter and spring, e.g. the Civil Aircraft for Regular Investigation of the Atmosphere Based on an Instrument Container (CARIBIC) in March 1999 and January 2001 (Papaspiropoulos et al., 2002), and the Indian Ocean Experiment (INDOEX) in February-March 1999 show elevated aerosol amounts near 8 – 12 km over the Indian Ocean and South Asia (De Reus et al., 2001). Recently, using a set of model simulations, Chavan et al., (2021) reported the transport of biomass burning aerosols to the upper troposphere by the convection in spring 2013.



Here, we investigate the source of the very large aerosol loading over the Arabian Sea
during spring. These aerosols produce atmospheric warming leading to enhanced water vapor
that is transported to the UTLS. Once in the lower stratosphere, the water vapour is
transported globally, which has implications for tropospheric temperatures and possibly
stratospheric ozone. For this purpose, we performed a series of five simulations using the
ECHAM6-HAMMOZ model for changes in anthropogenic aerosol over South Asia.
**2. Model simulations and satellite data**
**2.1 ECHAM6-HAMMOZ experimental set-up**
We used the state of art ECHAM6–HAM aerosol–chemistry-climate model. It
comprises of the general circulation module ECHAM6, coupled to the aerosol and cloud
microphysics module HAM (Stier et al., 2005; Tegen et al., 2019). HAM predicts the
nucleation, growth, evolution, and sinks of sulfate, black carbon (BC), particulate organic
matter (POM), sea salt (SS), and mineral dust (DU) aerosols. The size distribution of the
aerosol population is described by seven log-normal modes with prescribed variance as in
the M7 aerosol module (Stier et al., 2005; Zhang et al., 2012). Moreover, HAM explicitly
simulates the impact of aerosol species on cloud droplet and ice crystal formation. Aerosol
particles can act as cloud condensation nuclei or ice-nucleating particles. Other relevant
cloud microphysical processes such as evaporation of cloud droplets, sublimation of ice
crystals, ice crystal sedimentation, and detrainment of ice crystals from convective cloud
tops are simulated interactively (Neubauer et al., 2014). The anthropogenic and fire
emissions of sulfate, black carbon (BC), and organic carbon (OC) are based on the
AEROCOM-ACCMIP-II emission inventory. Other details of the model and emissions are
reported by Fadnavis et al. (2017a, 2019, 2021a, b).





The model simulations are performed at the T63 spectral resolution corresponding to
1.875°x1.875° in the horizontal dimension, while the vertical resolution is described by 47
hybrid σ−p levels from the surface up to 0.01 hPa (approx. 80 km). The simulations have
been carried out at a time step of 20 min. Monthly varying Atmospheric Model Inter-
comparison Project (AMIP) sea surface temperature (SST) and sea ice cover (SIC) (Taylor et
al., 2000) were used as lower boundary conditions.
We performed five model experiments: (1) a control (CTL) simulation where all aerosol
emissions are included and four perturbed experiments where  (2) all anthropogenic aerosol
emissions (black carbon, organic carbon, and sulfate) are switched off over South Asia (75 –
100° E, 8 – 40° N, see Fig. 1) during the study period (2001 – 2016) (referred to as Aerooff),
(3) only anthropogenic black carbon emissions (BC) switched off during the study period,
(BCoff), (4) only anthropogenic organic carbon (OC) emissions switched off (OCoff) during
the study period, and (5) only anthropogenic sulfate aerosol emissions switched off (Suloff)
during the study period (see Table 1). All simulations were performed from 1 January 2001 to
December 2016 from stabilized initial fields created after a model integration for one year.
Dust emission parameterization is the same in all the simulations and is based on Tegen et al.
(2002). The analysis is performed for spring (March – May) averaged for the period 2001 –
2016. We compare the CTL with Aerooff, BCoff, OCoff, and Suloff simulations to
understand the impact of south Asian anthropogenic aerosols over the Indian region and
surrounding ocean.





Table -1: Details of eECHAM6-HAMMOZ model simulations performed in this study.

| Experiment name | Duration | Aerosol species on/off | Boundary conditions |
|---|---|---|---|
| CTL | 2001 – 2016 | All aerosols species globally, as per AEROCOM-ACCMIP-II emission inventory. | AMIP Sea surface temperature and sea ice concentration. |
| Aerooff | 2001 – 2016 | Anthropogenic BC, OC, and sulfate aerosols switch off over South Asia during 2001 – 2016. | AMIP Sea surface temperature and sea ice concentration. |
| BCoff | 2001 – 2016 | Anthropogenic BC aerosols switch off over South Asia during 2001 – 2016. | AMIP Sea surface temperature and sea ice concentration. |
| OCoff | 2001 – 2016 | Anthropogenic OC aerosols switch off over South Asia during 2001 – 2016. | AMIP Sea surface temperature and sea ice concentration. |
| Suloff | 2001 – 2016 | Anthropogenic sulfate aerosols switch off over South Asia during 2001 – 2016. | AMIP Sea surface temperature and sea ice concentration. |


**2.2** AOD satellite observations
In this study we use the last fifteen years (2001 – 2016) of aerosol optical depth at 0.55 μm
(AOD) obtained from the Moderate Resolution Imaging Spectroradiometer
(MODIS) instrument onboard the NASA EOS Terra satellite. The MODIS instrument
measure radiance in 36 spectral channels at spatial resolution ranging from 250 m to 1 km
with a 2300 km wide swath, allowing for almost daily global coverage. Terra MODIS
(MOD08_M3 V6.1) AOD aerosol products are retrieved using the Deep Blue (DB) algorithm
(Mhawish et al., 2019). The algorithm calculates the column aerosol loading at 0.55 μm over
land and ocean. The AOD data from MODIS Terra can be downloaded from
https://ladsweb.modaps.eosdis.nasa.gov/archive/allData/61/MODATML2/


AOD data from the Multi-angle Imaging Spectro-Radiometer (MISR) for the same period as
MODIS (2001 – 2016) is also used for model evaluation. The MISR sensor onboard the Terra
satellite has been operational since 1999. It makes measurements at four spectral bands
centered at 443 nm, 555 nm, 670 nm, and 865 nm (Diner et al., 2008). In this study we used,
level 3 (MIL3MAE_v4) monthly mean aerosol optical depth at 555 nm wavelength at spatial
resolution   $0.5° \times 0.5°$.   The   MISR   AOD   data   is   available   for   download   at
https://misr.jpl.nasa.gov/getData/accessData/.

### 2.3  Model evaluation

We evaluate model performance by comparing simulated AOD (from CTL simulations)

with MISR and MODIS data for the spring season. The model simulations show high
amounts of AOD over the Indo-Gangetic plain, Myanmar, and East Asia, consistent with
MODIS and MISR observations, despite quantitative differences (Fig. 2). Compared to
observations, the model underestimates AOD over the Indo-Gangetic plain (model: 0.15 to
0.4, MODIS: 0.4 to 0.8, MISR: 0.3 to 0.5) and overestimates AOD over East Asia (model:
0.6 to 1.4, MODIS: 0.4 to 1.2, MISR: 0.2 to 0.5). Over the Myanmar region, the model
underestimates AOD in comparison to MODIS, but overestimates it in comparison to MISR
(model: 0.15 to 0.5, MODIS: 0.3 to 0.8 MISR: 0.15 to 0.3). There are differences among
satellite observations and between the model and observations. The differences are due to
uncertainties in the model due to model transport processes, emission inventory, and
parametrisations (Fadnavis et al. 2014, 2015, 2018, 2019) and there are uncertainties in
satellite measurements (Bibi et al., 2015). With model biases present in both the CTL and
the perturbed simulations, investigating anomalies removes some of the model bias. In the
past Fadnavis et al. (2018, 2019, 2020, 2021a,b) reported model evaluations for AOD,
absorbing aerosol index, precipitation, mixing ratio of black carbon aerosol and cloud ice





with various measurements that show the fair performance of the ECHAM6-HAMMOZ
model.

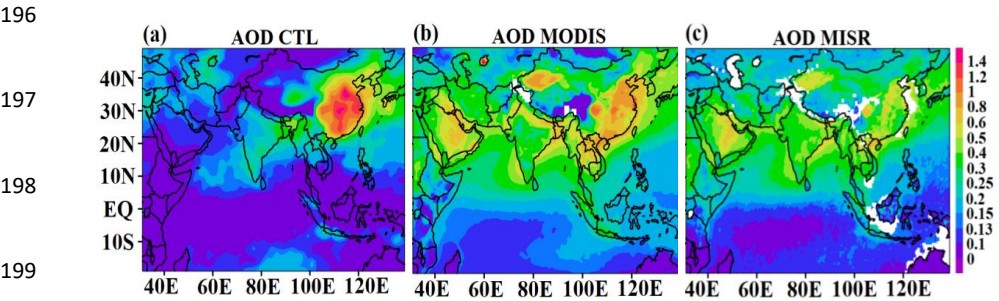




Figure 2: Spatial distribution of AOD average for the spring season during 2001 – 2016, from
(a) ECHAM6-HAMMOZ CTL simulations, (b) MODIS measurements average for the spring
season during 2001 – 2016, (c) MISR measurement average for the spring season during
2001 – 2016.
**4. Results and discussions**
**4.1 Transport of South Asian aerosols to the North Indian Ocean**
The spatial distribution of AOD anomalies from the Aerooff simulation shows positive
anomalies of AOD extending from South Asia to the Arabian Sea and the North Bay of
Bengal (10 - 20° N) (Fig. 3a). The wind vectors indicate that these are transported from the
Indo-Gangetic plain to the Arabian Sea and the Bay of Bengal. The transported aerosols
enhanced the AOD by 0.18 - 0.8 (30 - 80 %) over the North Bay of Bengal and by 0.02 - 0.12
(20 - 60 %) over the Arabian Sea. This is consistent with previous studies where 50 - 60 %
enhancements in the AOD over the tropical Indian Ocean due to anthropogenic aerosols have
been reported (Satheesh et al. 1999; Jose et al. 2020). Chemical analysis of aerosols observed
over the south-eastern coastal Arabian Sea also shows the dominance of anthropogenic
aerosols having sources over the Indian region (73 %) (Aswini et al., 2020). Analysis of
MODIS satellite observations (2003 – 2017) likewise shows that anthropogenic sources
contributed ~60 – 70% to the aerosol loading over the east coast and west coast of India (Jose



et al. 2020). Measurements over the Equatorial Indian Ocean further show a substantial
increase in AOD (~80 %) due to anthropogenic aerosols (Gogoi et al., 2019).
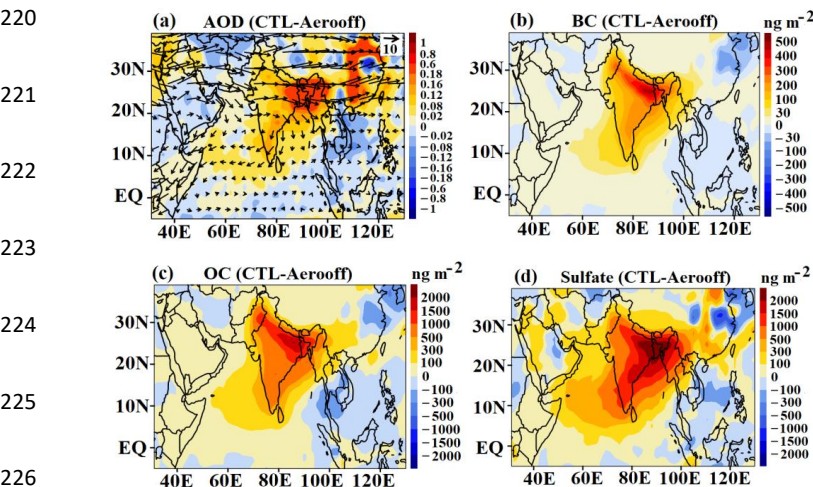
Figure 3: Spatial distribution of (a) AOD anomalies averaged for spring during 2001 –
2016 (CTL - Aerooff), and anomalies of tropospheric column of (b) BC, (c) OC, and (d)
sulfate aerosols (ng m$^{-2}$) (CTL-Aerooff). The vectors in Fig.1a indicate winds (m s$^{-1}$) at
850 hPa.
The distribution of anomalies in the tropospheric column of BC, OC, and sulfate aerosols also
indicates that these aerosols are transported from South Asia to the Bay of Bengal and the
Arabian Sea (Fig. 3b-d). Enhancement of sulfate and OC aerosol (100 – 2000 ng m$^{-2}$) is
higher than BC (100 – 500 ng m$^{-2}$) over the South Asian region (Fig. 3b-d). The total
carbonaceous aerosol (BC and OC together) dominates over the sulfate aerosols. These
anthropogenic aerosols over the tropical Indian Ocean affect the radiation budget and cloud
cover over the Indian Ocean (Satheesh et al. 1999; McFarquhar and Wang 2006).




## 3.2. Radiative forcing

The anthropogenic aerosols over the tropical Indian Ocean affect the radiation budget and cloud cover (McFarquhar and Wang 2006). Here, we discuss the impact of south Asian anthropogenic aerosols on radiative forcing (RF). Figures 4a-c show anomalies in net radiative forcing (RF) at the TOA, surface, and in-atmosphere (TOA - surface) for Aerooff simulations (CTL - Aerooff). The anthropogenic aerosols have produced a cooling at the TOA (except over the Indo-Gangetic plain) and surface (see Fig. 4a-b). The simulated RF values over the Arabian Sea (55 – 70° E, 8 – 20° N), Bay of Bengal (88 – 92° E, 12 – 20° N), and Indo-Gangetic Plain (75 – 83° E, 26 – 30° N) are tabulated in Table-S1. The RF estimates show that aerosols have produced significant cooling at the TOA and surface over the Arabian Sea (TOA: $-0.72\pm0.14$ W m$^{-2}$, surface:$-3.0\pm0.28$ W m$^{-2}$), Bay of Bengal (TOA:$-1.24\pm0.15$ Wm$^{-2}$, surface: $-5.14\pm0.44$ W m$^{-2}$), and in-atmospheric warming over the above regions (Arabian Sea $+2.27\pm0.19$ W m$^{-2}$; Bay of Bengal: $+3.89\pm0.30$ W m$^{-2}$) (Fig. 4 c). The Indo Gangetic Plain shows positive anomalies of RF at the TOA ($+1.27\pm0.16$ Wm$^{-2}$), negative at the surface ($-11.16\pm0.50$ Wm$^{-2}$), and atmospheric warming of $+12.44\pm0.42$ W m$^{-2}$. In agreement with our results, several previous studies have reported negative RF at the surface and TOA, and atmospheric warming over the north Indian Ocean caused by enhanced anthropogenic aerosol. For example, Pathak et al. (2020) reported negative aerosol RF at the TOA ($-2$ to $-4$ W m$^{-2}$) over the Bay of Bengal and the Arabian Sea during spring 2009 - 2013. Reddy et al., (2004) estimated positive in-atmosphere RF over the North Indian Ocean ($+25$ W m$^{-2}$). The aerosol radiative forcing estimated from satellite measurements (January to March 1999) over the north Indian ocean is also negative at TOA ($-4$ and $-14$ W m$^{-2}$) and surface ($-12$ to $-42$ W m$^{-2}$) (Satheesh and Ramanathan 2000; Rajeev and Ramanathan et al, 2001). The clear sky aerosol direct radiative forcing estimated from measurements during the INDOEX experiment (January to March in 1999) over the north Indian Ocean also show





similar results (TOA: -7 W m$^{-2}$, surface: -23 W m$^{-2}$, and in-atmosphere: +16 W m$^{-2}$)
(Ramanathan et al. 2001). These studies attribute positive in-atmospheric radiative forcing to
absorbing aerosols (especially black carbon) that lead to heating of the atmosphere (Rajeev
and Ramanathan 2001; Satheesh et al 2002).

Analysis of the perturbed model experiments indicates that anthropogenic BC

aerosols (Fig. 4d-f) have produced a warming at the TOA (Arabian Sea: 1.24±0.13 W m$^{-2}$,
Bay of Bengal: 1.54±0.26 W m$^{-2}$, Indo-Gangetic Plain: 4.33±0.17 W m$^{-2}$) and cooling at the
surface (Arabian Sea: -2.56±0.25 W m$^{-2}$, Bay of Bengal: -3.70±0.49 W m$^{-2}$, Indo-Gangetic
Plain:-9.27±0.37 W m$^{-2}$). OC (Fig. 4g-i) and sulfate (Fig. 4j-l) aerosols have produced
significant cooling at the TOA (OC: -0.21±0.13 to -0.44±0.15 W m$^{-2}$; Sulfate: -1.55±0.16 to -
2.14±0.17 W m$^{-2}$) and surface (OC: -0.49±0.31 to -2.56±0.45 W m$^{-2}$, Sulfate: -1.19±0.24 to -
2.67±0.36 W m$^{-2}$) over the above regions (listed in Table-S1).  Figures 4d, 4g, and Fig. 4j
further confirm our finding that the positive anomalies of radiative forcing in the Indo-
Gangetic plain are due to BC aerosols because of its absorbing property. All the aerosols
produce in-atmospheric warming over the Indian region and the north Indian Ocean. The
atmospheric warming over the Arabian Sea and Bay of Bengal is due to BC and OC aerosols
with larger contributions by the BC aerosols.








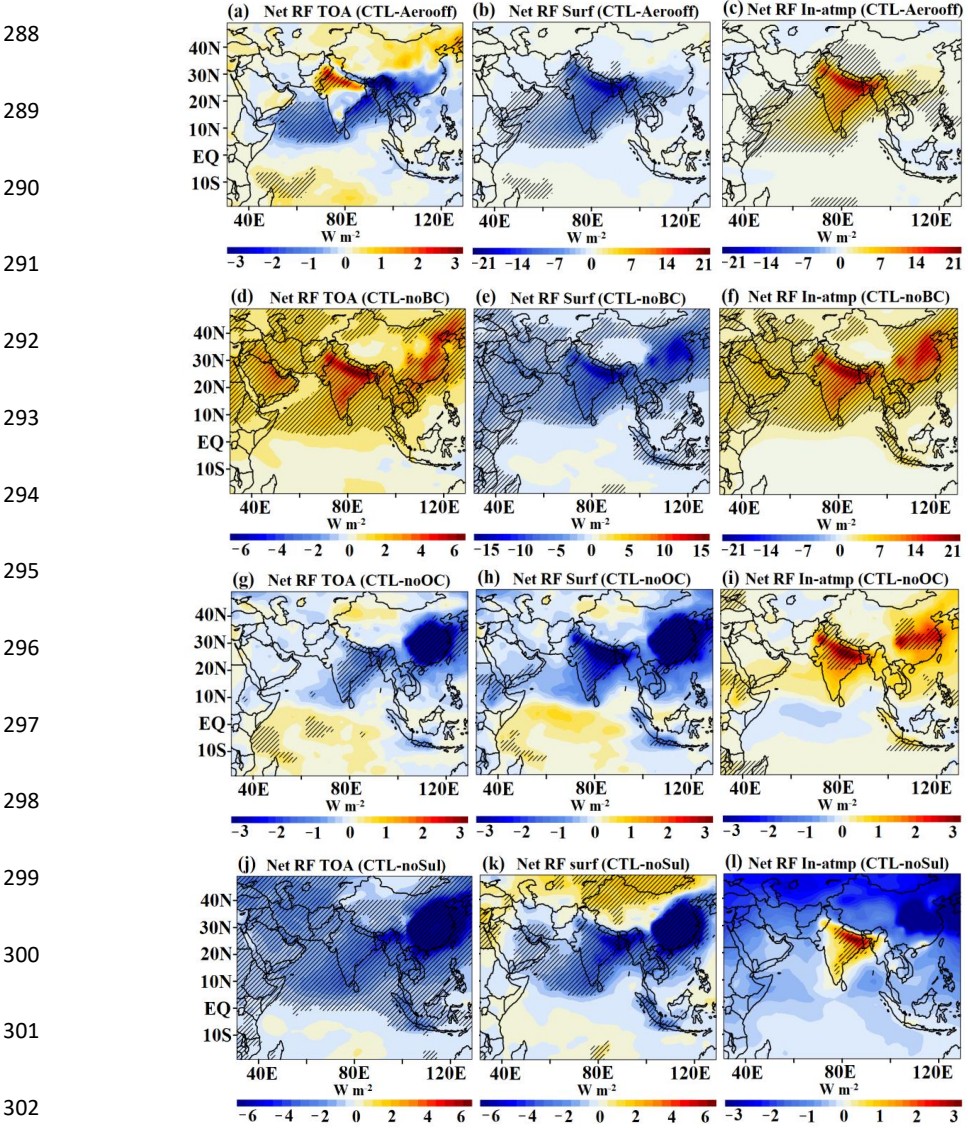

Figure 4: Spatial distribution of net aerosol radiative forcing (CTL - Aerooff) (W m$^{-2}$ ) averaged for spring during 2001 – 2016 (a) TOA, (b) same as (a) but for surface, (c) same as (a) but for in-atmosphere (TOA - surface), (d) spatial distribution of radiative forcing at the TOA (CTL – BCoff) averaged for spring during 2001 – 2016, (e) same as (d) but for surface, (f) same as (d) but for in-atmosphere (TOA - surface), (g) spatial distribution of radiative forcing at the TOA (CTL - OCoff) averaged for spring during 2001 – 2016, (h) same as (g) but for surface, (i) same as (h) but for in-atmosphere (TOA - surface), (j) spatial distribution of radiative forcing at the TOA (CTL - Suloff) averaged for spring during 2001 – 2016, (k) same as (j) but for surface, (l) same as (k) but for in-atmosphere (TOA - surface). The hatched lines in figure a-l indicate 99% confidence level for the mean differences.



### 3.3. Transport of Asian anthropogenic aerosols into the UTLS

Further, we investigate the vertical distribution of aerosols that are transported to the north Indian Ocean. Meridional sections over the Arabian Sea (60 – 75° E) and Bay-of-Bengal (75 – 95 °E) for BC, OC, and sulfate aerosol anomalies indicate that these aerosols are transported from the boundary layer of both regions and north India into the UTLS (Figure 5 and Fig. S1). The spring convection occurring over the Arabian Sea and Bay-of-Bengal which is shown by the combined distribution of CDNC and ICNC (Fig. S2a) plays an important role in the vertical transport. The prevailing spring convection is further invigorated over the Arabian Sea by the transported aerosols there which is not the case for the Bay of Bengal region (Fig. S2b). The aerosol loading over the North-Indian region forms clouds and elicit convection there (Fig. S2c-d). The distribution of wind resolved circulation shows a strong ascent over the Arabian Sea, and the Bay-of-Bengal regions, while the steep orography of the Himalayas over North India also plays an important role in the vertical transport to the upper troposphere (Fig. 5 and Fig. S1). Figure 5 also shows that aerosols induce a secondary circulation, ascending winds over 10 – 30° N and descent over 30 – 40° S. BC, OC, and sulfate aerosols are transported to the UTLS, moving southward and downward ~30 – 40° S (Fig. 5a-f, and Fig. S1) due to this secondary circulation. The aerosol enhancement in the lower troposphere (1000 – 500 hPa) over 30 – 40° S is therefore due to the combined impact of horizontal transport and downward transport from the UTLS due to this secondary circulation. Further, in the UTLS these aerosols are transported globally. There is enhancement in the Arctic (BC: 0.6 to 1.5 ng m$^{-3}$, OC: 0.4 to 7 ng m$^{-3}$, Sulfate: 0.1 to 20 ng m$^{-3}$) and Antarctic (BC: 0.6 to 3 ng m$^{-3}$, OC: 1 to 5 ng m$^{-3}$, Sulfate: 6 to 40 ng m$^{-3}$) in the lower stratosphere (180 – 90 hPa) (see Figure 5). Our analysis shows that transport to the Arctic and Antarctic occurs every year in the UTLS which causes heating in the lower stratosphere (see Section 3.4).





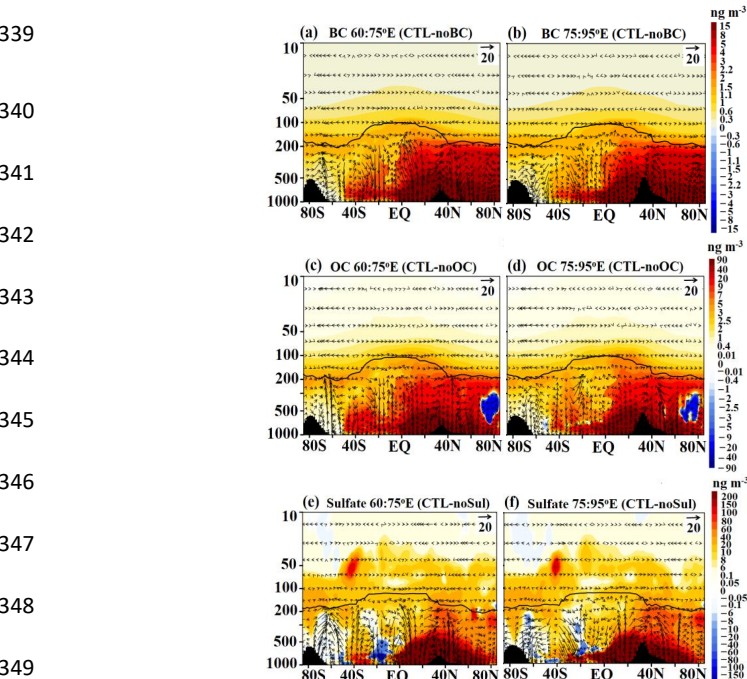








Figure 5: Meridional cross-section over Arabian Sea (averaged 60 – 75° E) and Bay-of-
Bengal (75 – 95° E) and for the spring season during 2001 – 2016 of anomalies (ng m$^{-3}$) of
(a–b) BC aerosols (CTL-BCoff), (c–d) OC aerosols (CTL-OCoff), (e–f) sulfate aerosols
(CTL-Suloff). Vectors in Figs. a-f indicate anomalies of winds (m s$^{-1}$) (the vertical velocity
field has been scaled by 300 and the units are m s$^{-1}$).

**3.4. Impacts on the heating rate and water vapor**
Carbonaceous aerosols absorb solar radiation, leading to atmospheric heating, while
predominately scattering aerosols such as sulfate reflect and scatter back solar radiation,
therefore cooling the atmosphere below (Fadnavis et al., 2019). The vertical distribution of
heating rate anomalies induced by all the anthropogenic Asian aerosols (CTL - Aerooff) over
the North Indian Ocean (Arabian Sea and North Bay of Bengal, 50 – 100° E) indicates a
significant increase in heating rates in the region of elevated anthropogenic aerosols in the
troposphere (0.05 K d$^{-1}$) and stratosphere (0.002 K d$^{-1}$) (Fig. 6a-d, Fig. 5, and Fig. S1).
Heating rate anomalies estimated over the North Indian Ocean from BC (CTL - BCoff), OC



(CTL - OCoff), and Sulfate (CTL - Suloff) show that BC and  OC aerosols produce heating in
the troposphere (10 – 40º N) (BC: 0.001 to 0.05 K d$^{-1}$, OC: 0.0002 to 0.02 K d$^{-1}$) and
stratosphere (100 – 50 hPa, 90ºS – 90ºN)  (BC: 0.001 to 0.008 K d$^{-1}$, OC: 0.0002 to 0.002 K
d$^{-1}$), while sulfate aerosols produce atmospheric cooling in the troposphere -0.001 to -0.05
(500 – tropopause) and stratosphere -0.001 to -0.008 K d$^{-1}$ (tropopause – 50 hPa)  (Fig. 6a-d).
There is anomalous heating in the tropical stratosphere (20° S – 20° N) (0.001 to 0.002 K d$^{-1}$)
seen in CTL-Aerooff simulations (Fig. 6a), mainly due to carbonaceous aerosols (Fig. 6b-c).

The zonal mean distribution of heating rates (Fig 6 e-h) shows that the South Asian

carbonaceous aerosols lead to 0.002 – 0.01 K d$^{-1}$ heating in the lower stratosphere globally
(100 – 50 hPa) (Fig. 6f-g), larger than the cooling induced by sulfate aerosols (Fig. 6h).

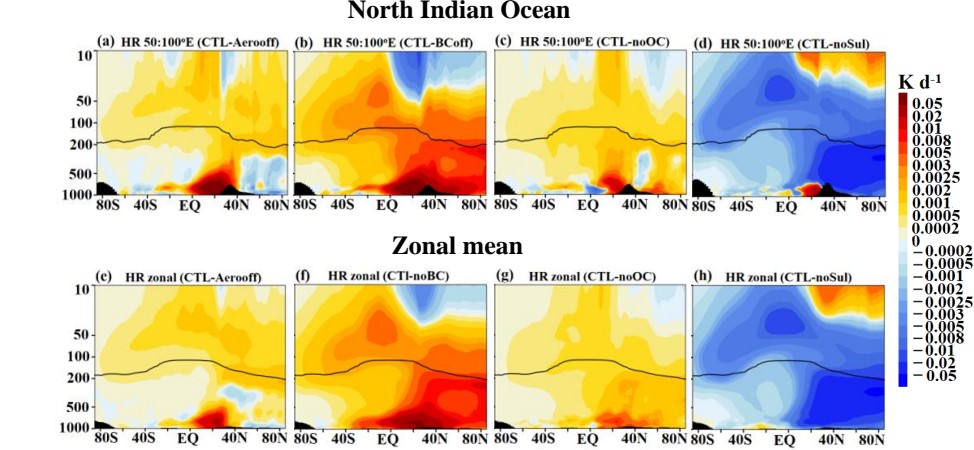

Figure 6: Meridional cross-section over the North Indian Ocean (averaged over the
Arabian Sea and Bay of Bengal region, 50 – 100° E) of anomalies of heating rates (K d$^{-1}$)
averaged for the spring season during 2001 – 2016 (a) from CTL - Aerooff simulation, (b)
same (a) but from CTL - BCoff simulation (c) same (a) but from CTL -OCoff simulation,
(d) same (a) but from CTL - Suloff simulation. (e) Zonal mean (0 –360°) anomalies in
heating rate for CTL - Aerooff simulation, (f) same as (e) but from CTL - BCoff
simulation, (g) same as (e) but from CTL - OCoff simulation, (h) same as (e) but from
CTL - Suloff simulation.



In general, these aerosols increase heating in the troposphere over the South Asian region (Fig. 6a) and northern Arabian Sea and Bay of Bengal (10 – 30° N). This enhanced heating invigorates the convection process, which results in an increase in cloud cover (Fig. S2c) and deepening of the OLR (Fig. S2d). The invigorated convection provides positive feedback on the vertical ascent into the free troposphere that extends to above the tropopause into the lower stratosphere over the Arabian Sea and Bay-of-Bengal-North-India region (Fig. S3a-b) (Fadnavis et al., 2013; Randel et al., 2010).

The vertical distribution of water vapor over the Indian Ocean (CTL - Aerooff) shows that water vapor (0 – 0.3 ppmv) is transported to the UTLS from the Arabian Sea (55 – 70° E, 0 – 30° N) (Fig. 7a) along the path of elevated aerosols (Fig. 5). Interestingly, there is an enhancement in water vapor over the southern Indian Ocean (20 – 30° S, 55 – 70° E) along the path of the descending branch of aerosols (BC, OC, and sulfate). This is due to the significant heating caused by carbonaceous aerosols (Fig. 6b-c) which leads to enhancement of tropospheric water vapor (Fig. 7a) over the Arabian Sea. The zonal mean (averaged for 0 – 360°) anomalies of water vapor (Fig. 7b) show an enhancement by 0.03-0.08 ppmv (0 – 4 %) in the global stratosphere (Fig. 7b). There is an enhancement in the lower stratosphere in the Antarctic (60 – 90º S) by 0.01 to 0.03 ppmv and in the Arctic (80 – 90º N) by 0.01 - 0.1 ppmv caused by anthropogenic aerosols (CTL-Aerooff).

The impact of BC (CTL - BCoff), OC (CTL - OCoff), and Sulfate (CTL - Suloff) on water vapor distribution (Figs. 7 c-f) shows that BC aerosols play a major role in water vapor enhancement in the stratosphere  (Fig. 7 c), (100 – 10 hPa). Water vapor enhancement by BC aerosols over the Arabian Sea region is ~0.03 – 0.3 ppmv (Fig. 7c) and 0.01 – 0.2 ppmv in the global stratosphere (Fig. 7d). There is significant enhancement of water vapor due to BC aerosols in the Arctic (0.01 – 0.2 ppmv) and Antarctic (0.01 – 0.1 ppmv) (Fig. 7d).  The water vapor enhancement by OC aerosols is negligible in the tropical stratosphere and there is no





contribution of sulfate aerosols (Fig. 7 e-f). The sulfate aerosols cause negligible heating by
abortion of infra-red radiations over the Arabian Sea that leads to water vapor enhancement
from the boundary layer to the mid-troposphere (500 hPa), near the tropopause, and in the
path of descending branch of secondary circulation over the South Indian Ocean (~20° S)
(Fig. 7f).

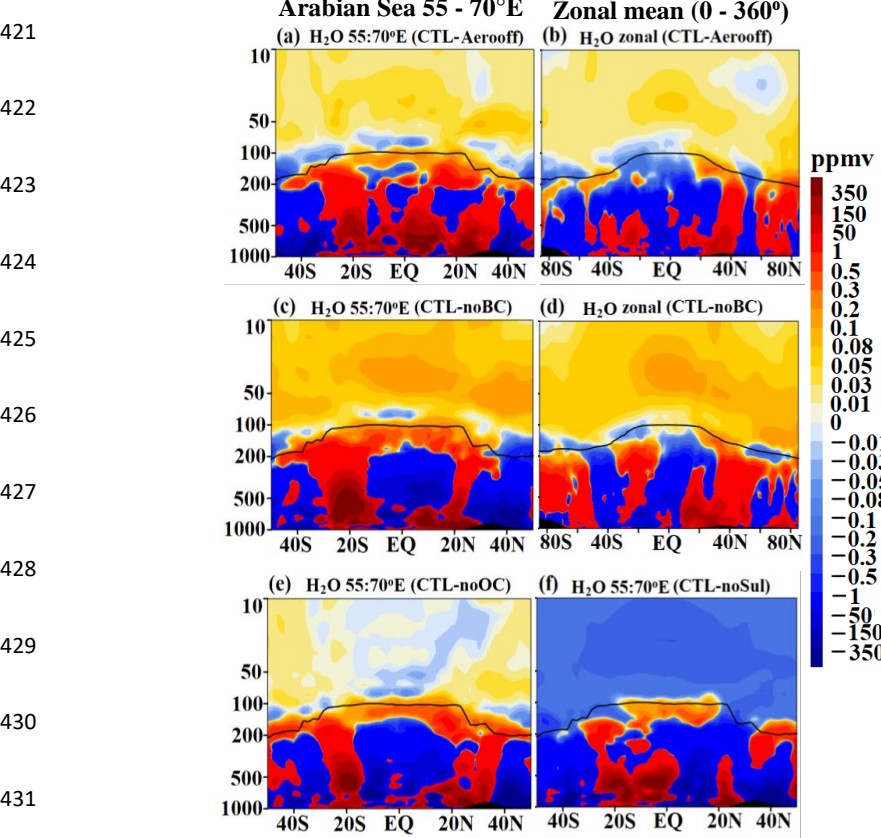

Figure 7: (a) Meridional cross-section over Arabian Sea region (averaged 55 – 70° E) of
anomalies of water vapour (ppmv) (CTL - Aerooff) the for spring season during 2001 – 2016,
(b) same as (a) but zonal mean (average for longitudes 0 – 360°), (c) same as (a) but from
CTL - BCoff simulations, (d) same as (c) but zonal mean (average for longitudes 0 –360°),
(e) same as (a) but from CTL - OCoff simulations, (f) same as (a) but from CTL - Suloff
simulations.
Although the focus of the manuscript is on transport of aerosols during the spring season, it
should be noted that the anthropogenic South Asian aerosols are also transported to the UTLS



during the monsoon season (Fadnavis et al., 2013, 2017, 2019). Annual distribution
anomalies of aerosols (average of BC, OC and sulfate) show transport of aerosols into the
UTLS during spring and monsoon season (April to September) from the Arabian Sea, Bay-
of-Bengal-North-India region (Fig. 8a). These aerosols enhance tropospheric heating thereby
transporting elevated water vapour into the lower stratosphere (Fig. 8b). Injection of aerosols
into the lower stratosphere occurs every year however there is interannual variability. We
show the vertical distribution of aerosols for two normal years when there was no large scale
ocean-atmosphere coupling phenomenon like El Niño southern oscillation or Indian Ocean
Dipole (2008 and 2016) in Fig. S4a-b. It also shows transport of aerosols into the lower
stratosphere during spring and monsoon seasons (March-September). The aerosol induced
enhanced water vapour also shows enhancement in the lower stratosphere during the same
time (Figs. S4c-d). In the lower stratosphere, these aerosols persist for a few months (Fig. 8a)
thus their effect will be seen for an extended time.

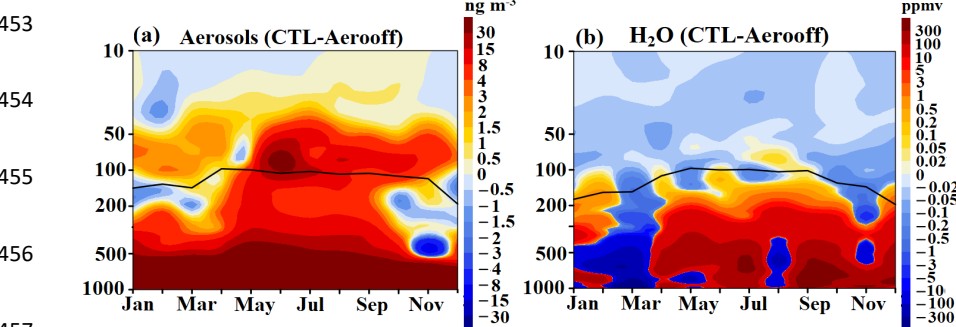

Figure 8: (a) Annual distribution of anomalies of aerosols (CTL - Aerooff) (averaged of BC, OC and sulfate aerosols) (ng m$^{-3}$) averaged for Arabian Sea, Bay-of-Bengal-North-India region (55 – 110° E, 10 – 30° N), (b) same as (a) but for water vapour (ppmv).

Further, we analyze the correlation between heating rates and carbonaceous aerosol amounts

in the UTLS (180 – 70 hPa) in the Arctic and Antarctic during 2001 – 2016 (spring mean)

(Fig. 9) from Aerooff, BCoff, and OCoff in comparison with CTL simulations. The





carbonaceous aerosols show a positive correlation (correlation coefficient r: 0.57 to 0.94)
with the UTLS heating rates indicating that transported carbonaceous aerosols enhance UTLS
heating in the Arctic and Antarctic. It should be noted that transport of aerosols to the Arctic
and Antarctic also occurs during the monsoon season (Fadnavis et al., 2017a, 2017b, 2019)
which may affect the dynamics and aerosol amounts in the spring of the next year in the
UTLS.

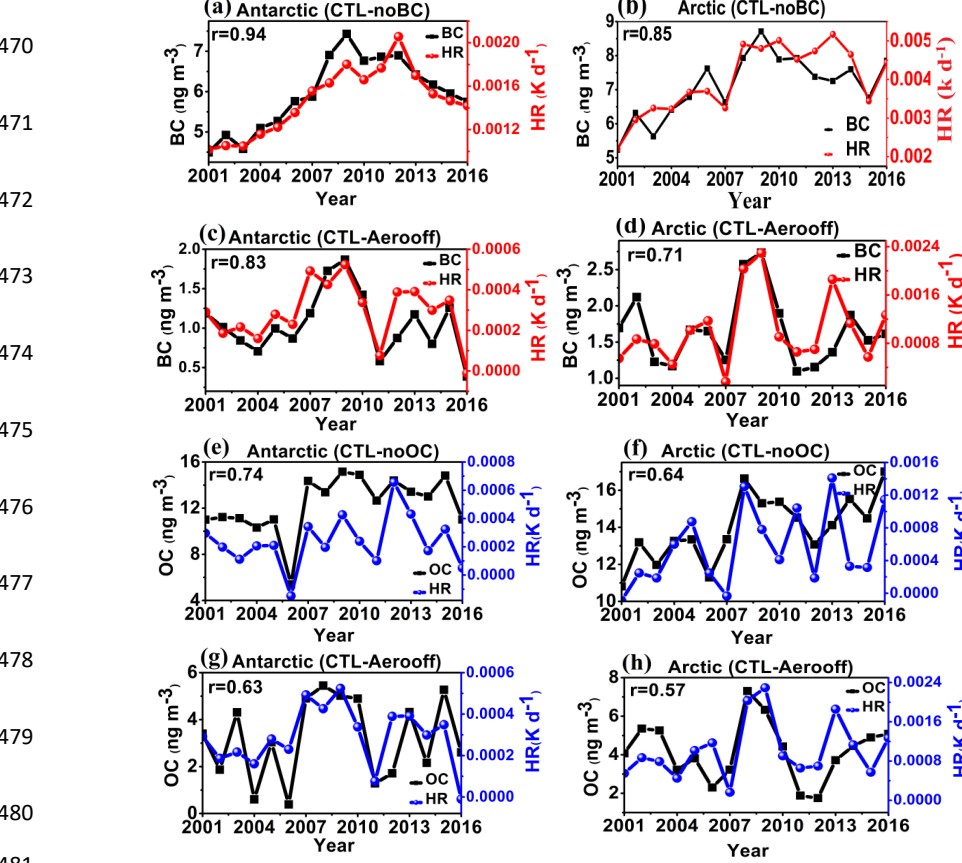

Figure 9: (a) Time series of BC aerosols and heating rates averaged for spring and UTLS (180 – 70 hPa) at the Antarctic (60 – 90 °S, 0 – 360 °)(from CTL - BCoff, (b) same as (a) but in the Arctic (65 – 85° N, 0 – 360°), (c) same as (a) but from CTL - Aerooff, (d) same as (b) but from CTL - Aerooff, (e) same as (a) but for OC (CTL - OCoff), (f) same as (b) but for OC (CTL - OCoff), (g) same as (c) but for OC, (h) same as (d) but for OC. The correlation coefficient (r) between anomalies of BC/OC aerosols and heating rates is indicated in panels a-h.





Importantly, South Asian aerosols enhance water vapor in the lower stratosphere, globally.
Water vapor being a greenhouse gas further enhances the heating of the troposphere leading
to a positive feedback. The increase in water vapor in the stratosphere also warms the Earth's
surface (Shindell, 2001; Solomon et al., 2010). Solomon et al. (2010) estimated that an
increase in the stratospheric water vapor by 1 ppmv accounts for 0.24 W m$^{-2}$ radiative
forcing. The SABER and MLS observations showed an increase in stratospheric water vapor
by 0.45 ppmv globally during 2003 – 2017 (Yue et al., 2019). Thus the radiative forcing due
to water vapor increase (0.02 – 0.14 ppmv) in response to South Asian anthropogenic
aerosols is not negligible for surface warming globally. Further, increasing stratospheric
water vapour could also lead to ozone depletion (e.g., Shindell, 2001, Robrecht et al., 2019).
**4. Conclusions**
A series of ECHAM6-HAMMOZ chemistry-climate simulations for South Asian
anthropogenic aerosols were used to understand the transport pathways of Asian aerosols and
their associated impacts. The analysis is performed for the spring season, when emissions of
anthropogenic aerosols (BC, OC, and sulfate) over south Asia peak. The model simulations
show that large amounts of South Asian aerosols are transported during spring to the Arabian
Sea (increases in AOD by: 0.02 – 0.12 from CTL - Aero0ff) and Bay of Bengal (increases in
AOD by: 0.16 to 0.8 from CTL - Aerooff). The anthropogenic aerosols are further lifted up
into the UTLS from the Arabian Sea and Bay-of-Bengal-North-India. In the UTLS, they are
also transported to the southern hemisphere (30 – 40 °S) and downward in the troposphere by
the secondary circulation induced by the aerosol changes. In the UTLS, these aerosols (BC,
OC, and Sulfate) are transported globally.
The anthropogenic aerosol produces significant radiative impacts over the Indo-Gangetic
Plain (RF anomalies estimated from CTL-Aerooff simulations, TOA: +1.27±0.16 W m$^{-2}$,


Surface: -11.16±0.50 W m$^{-2}$, In-atmosphere: +12.44±0.42 W m$^{-2}$) and the Arabian Sea (RF at
the TOA: -0.72±0.14 W m$^{-2}$, surface: -3.00±0.28 W m$^{-2}$, In-atmosphere: +2.27±0.19 W m$^{-2}$).
Interestingly, RF at the TOA over the Indo-Gangetic Plain is positive +4.33±0.17 W m$^{-2}$ due
to the emission of BC aerosols. The anthropogenic aerosols enhance heating in the
troposphere (estimated from CTL-Aerooff) by 0.002 to 0.05 K d$^{-1}$ and UTLS by 0.001 to
0.002 K d$^{-1}$ leading to more cloud formation (cloud cover anomalies enhanced by 2 to 12 %)
and intensification of convection (OLR anomalies -0.5 to -10 W m$^{-2}$). This invigorated
convection provides a positive feedback on the vertical updraft of aerosols into the free
troposphere and above the tropopause into the lower stratosphere (Fadnavis et al., 2013;
Randel et al., 2010). The tropospheric heating/cooling caused by the anthropogenic aerosols
over South Asia and the North Indian Ocean during spring has implications on the Indian
summer monsoon rainfall a few months later (Fadnavis et al., 2017a; Fadnavis and
Chattopadhyay, 2017).

The heating of the troposphere by the carbonaceous aerosol increases evaporation and
thereby tropospheric water vapor amounts over the North Indian Ocean and adjoining
regions. The elevated water vapor is transported from highly convective regions (i.e. Arabian
Sea) to the UTLS, from where it is then transported globally. BC aerosols play a major role in
water vapor enhancement in the lower stratosphere, globally (increased water vapor by 0.01
to 0.1 ppmv). Water vapor being a greenhouse gas further enhances the heating of the
troposphere leading to a positive feedback. The increase in water vapor in the stratosphere
also warms the Earth's surface (Shindell, 2001; Solomon et al., 2010).
*Acknowledgments*: The authors thank the staff of the High Power Computing Centre (HPC)
in the Indian Institute of Tropical Meteorology, Pune, India, Pune, India.



**Data availability:** The data used in this study are generated from ECHAM6-HAMMOZ model simulations at the High-performance computing system in the Indian Institute of Tropical Meteorology, Pune, India. The AOD data from MODIS Terra used here can be downloaded from https://ladsweb.modaps.eosdis.nasa.gov/archive/allData/61/MODATML2/, and MISR from https://misr.jpl.nasa.gov/getData/accessData/.

**Author contributions**: S. F. initiated the idea. A. J., S. S., A. A., performed model analysis. R. M., and A. R. contributed to analysis and study design. All authors contributed to the writing and discussions of the manuscript.

**Competing Interests**: Some authors are members of the editorial board of Atmospheric Chemistry and Physics. The peer-review process was guided by an independent editor, and the authors have also no other competing interests to declare.



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
