# Peer review of "Tropospheric warming over the North Indian Ocean caused by the South Asian 1 anthropogenic aerosols: possible impact on the upper troposphere and lower 2 stratosphere 3 4 Suvarna Fadnavis1\*, Prashant Chavan1, Akash Joshi2, Sunil Sonbawne1, Asutosh Acharya"

_Atmospheric Chemistry and Physics, 2021_

## Referee Comment (RC2)

[referee-annotated manuscript omitted]

---

## Author Comment (AC1)

**Total heating rates (LW+SW)**

[Figure]

**Short wave heating rates**

**Long wave heating rates**

Figure: Zonal section (averaged $30^o$ E – $140^o$ E and for spring) of net heating rate (LW+SW), (Kmonth$^-$1) from simulations (a) CTL-Aerooff, (b) CTL-noBC, (c)CTL-noOC, (d) CTL-nosul, (e-h) same as (a-d) but for SW heating rates, (i-l) same as (a-d) but for LW heating rates.

---

## Author Response (AR1)

Fadnavis et al. present simulations with the chemistry-climate model ECHAM6-HAMMOZ to investigate the transport pathways and impacts of anthropogenic aerosols during spring. They perform five model simulations, one control run and four sensitivity runs. From their simulation results, they find that the carbonaceous aerosols cause an increase in heating rates and an increase in water vapor.

Reply: We thank the reviewer for their thorough assessment of our study and the valuable suggestions to improve the manuscript. We have now incorporated all suggestions in the manuscript at the line numbers mentioned in the replies below. The changes are also indicated in blue color in the revised manuscript.

**General comments:**

(1) The manuscript is generally well written and structured, but leaves several questions open. For example, the title states "possible implications". What exactly do you mean with that? The society, the atmosphere or a specific process? This is not answered throughout the paper (or if it is answered the message does not come across).

Reply (1): Thank you for pointing this out. We have changed the title of the manuscript as "Tropospheric warming over the North Indian Ocean caused by the South Asian anthropogenic aerosols: possible impact on the upper troposphere and lower stratosphere"

(2) I have difficulties to understand the connection between the heating rates and the increases in water vapour as well as the connection between aerosols and convection. Both seems to be essential for this study, but the underlying processes are not really explained. Thus, this needs definitely to be elaborated in more detail.

Reply (2): To highlight this connection more clearly, we have now performed an additional analysis on isentropic levels. The transport pathways of aerosol to the Southern Hemisphere are now therefore better illustrated. Our analysis shows that South Asian aerosols are transported to the North Indian Ocean. These aerosols are then lifted into the upper troposphere and lower stratosphere (UTLS) by the ascending branch of Hadley circulation. In the UTLS, they enter the westerly jet. They are further transported to the Southern Hemisphere (~15º S – 30º S), and downward (320 – 340K) via an equatorial Atlantic westerly duct (5º S – 5º N, 10º W-40º W) and shifted Pacific westerly duct (5º S – 5º N, 95º E – 140º E). The shifting of a Pacific westerly duct may be due to higher Rossby wave breaking caused by the South Asian aerosols. (See discussion in section 3.3)

The carbonaceous aerosols (mainly BC) produce short wave heating along the transport pathway. This leads to an increase in temperature and therefore water vapour concentrations. We have now removed the discussions on the connection between aerosols and convection. (See discussion in section 3.4).

(3) Additionally, I have the feeling that the connections between these specific processes and the according changes in heating and water vapour are overrated. Since the numbers are quite low.

Reply(3): Since the gradient of water vapor is very large between troposphere and stratosphere we have expressed it in percentage change. Our analysis shows an enhancement in water vapor by 1 - 10 % in the UTLS over tropical and sub-tropical latitudes (45º S – 45º N). While we agree with the reviewer that the numbers are not extremely large, a change in UTLS water vapor of 10% is not negligible. (L39-40 and section 3.4)

(4)    Are the AODs only used for model evaluation? If yes, this could be provided in the supplement. How good are the heating rates, aerosol distributions etc. simulated in the model? Are the model simulations reliable? Wouldn't it be worth to generally compare the model simulations with observations?

Reply (4): We agree with the reviewer's point and, as suggested, we have now moved the AOD analysis to Supplementary Material. We have also expanded the discussions on the comparison of observations with the model as follows:

"The model simulations show high amounts of AOD over the Indo-Gangetic plain (25º N – 35º N, 75º – 95º E), consistent with MODIS and MISR observations, despite quantitative differences (Fig. S1). Compared to observations, the model underestimates AOD over the Indo-Gangetic plain by ~18 % than MODIS and overestimates by 14% than MISR. While it underestimates over central India (15º – 24º N, 75º – 82º E) by 20 – 23 % compared to MODIS and MISR. There are differences among satellite observations and between the model and observations. The differences are due to (1) uncertainties in the model transport processes, the emission inventory, and the parameterizations. (Fadnavis et al. 2014, 2015, 2018, 2019) and (2) there are uncertainties in the satellite measurements (Bibi et al., 2015). The comparison of AOD from Aerosol Robotic Network (AERONET), MODIS and MISR show error of 0.03 to 0.05 (Kahn et al., 2007). With model biases present in both the CTL and the perturbed simulations, investigating anomalies removes some of the model bias." (see section S2, L36-48)

**Specific comments:**

(5) Title: "possible implications" what do you mean with that? Implications on what? The atmosphere? The society? This is also not at all explained/discussed throughout the paper.

Reply (5) Thank you for pointing this out. We have changed the title of the manuscript to " Tropospheric warming over the North Indian Ocean caused by the South Asian anthropogenic aerosols: possible impact on the upper troposphere and lower stratosphere"

(6)    P2, L32 and 35: The numbers alone are not helpful without any further explanations. Are these changes severe or negligible?

Reply(6): Heating rates are small since they are expressed per day. Now we have changed them to Kmonth$^{-1}$. We have now added percentage changes of heating rates to show their impact. (See L36-L37 and section 3.4).

(7)     P2, L37: Please clarify and state more precisely if these increases in heating are due to heating or due to transport.

Reply(7): The above sentence is re-written as "The increase in tropospheric heating due to aerosols results in an increase in water vapor concentrations, which are then transported from the North Indian Ocean-Western Pacific to the UTLS over 45º S – 45º N (increasing water vapor by 1 – 10 %), L37-40.

(8)     P4, L70: Are the here mentioned increases based on observations or model simulations? Please add.

Reply (8): It is from AEROCOM-ACCMIP-II emission inventory. It is stated at L70-L71.

(9)     P5, Figure 1: Where is the data shown coming from? Inventory, model simulation or observation?

Reply(9): It is from AEROCOM-ACCMIP-II emission inventory. It is mentioned at L100.

(10)    P6, L115: Add here some references. There are some studies that have investigated the impact of increases in water vapour on the polar stratosphere (e.g. Khosrawi et al., 2016; Thölix et al. 2016; Thölix et al., 2018).

Reply(10): We have repeated the analysis on isentropic levels. The new figures show water vapor enhancement in the UTLS between 45º S – 45º N. Hence, we could not cite the above references.

(11)    P6, L116: Add some sentences on the structure on the paper and what you are going to do in this paper? Especially since you keep this kind of information quite short in the abstract and introduction it would be worth to give here some more details.

Reply (11): As suggested we have added an outline of the paper at L119-122 as "The paper is structured as follows: the ECHAM6-HAMMOZ model simulations are provided in section 2, in section 3 we discuss the results on the transport of South Asian aerosols to the North Indian Ocean, radiative forcing, transport into the UTLS, and associated impacts on heating rates, while conclusions are summarised in section 4."

(12)    P6, L124: Why seven? Either skip this information or provide more details.

Reply(12): It is provided at L130-132 as "Nucleation mode, soluble and insoluble Aitken, soluble and insoluble Accumulation and soluble and insoluble Coarse modes.

(13)    P6, L125: Same here. What does M7 mean? Either skip this information or provide more details on what this module does or is.

Reply (13): It is deleted now.

(14)    P7, L140ff: Here definitely a motivation for your experiments is missing. Why are you switching of the aerosols? What kind of insights you can get from each simulation experiment?

Reply (14): The motivation is to understand (1) the transport pathways of South Asian anthropogenic aerosols, and (2) their impact over the Indian region, and UTLS (340K - 400K). It is mentioned at L160-161.

(15)    P9, L170: That you use the satellite data sets for model evaluation should be mentioned already in the beginning of the section.

Reply (15): It is moved to supplement as suggested in a comment (4).

(16)    P9, L182: Is AOD a unitless number? What is the typical range of AOD? What do these numbers tell me? How strong is the over-/underestimation? Something the reader should worry about? I would suggest to add differences in percent so that it is easier to follow if this is a large or small over-/underestimation.

Reply (16): Yes AOD is a unitless number. We have added values in percentages and discussion given below:

"The model simulations show high amounts of AOD over the Indo-Gangetic plain (25º - 35 ºN, 75º - 95º E), consistent with MODIS and MISR observations, despite quantitative differences (Fig. S1). Compared to observations, the model underestimates AOD over the Indo-Gangetic plain by ~18 % than MODIS and overestimates by 14% than MISR. While it underestimates over central India (15º – 24º N, 75º - 82º E) by 20 – 23 % compared to MODIS and MISR. There are differences among satellite observations and between the model and observations. The differences are due to (1) uncertainties in the model transport processes, the emission inventory, and the parameterizations. (Fadnavis et al. 2014, 2015, 2018, 2019) and (2) there are uncertainties in the satellite measurements (Bibi et al., 2015). The comparison of AOD from Aerosol Robotic Network (AERONET), MODIS and MISR show error of 0.03 to 0.05 (Kahn et al., 2007). With model biases present in both the CTL and the perturbed simulations, investigating anomalies removes some of the model bias." (see section S2, L36-48)

(17)    P9, L190: Give more information on the uncertainties of the satellite data. I guess there are validation studies that provide some uncertainty estimations so that you could provide some numbers for errors or biases of the satellite data.

Reply (17): We have now added numbers for errors of the satellite data as mentioned in Reply (16) (section S2, L45-47).

(18)    P10, L195: How reliable is the ECHAM6-HAMMOZ simulation? Is the model good enough for the here anticipated study?

Reply (18): We thanks for pointing this out. The ECHAM6-HAMMOZ model has previously been evaluated in several studies in the past, indicating its suitability for studies of this type. This is now more clearly mentioned in section S2, L48-51. Furthermore, we provide a comparison with model simulations (see Reply (14))

(19)    P10, Figure 2: The differences between model and observations are quite large. For me it looks like that there are serious problems in deriving AOD correctly in the model. Could provide this figure once again changing the scale of the mode so that it is possible to check if there is a qualitative agreement (thus to see if the model at least gets the AOD generally right).

Reply(19): We have changed the color scale in Figure-2 and zoom-in over the South Asian region. The revised figure shows qualitative agreement of the model with MODIS and MISR. As mentioned in Reply (16) differences in model are -23 to 18 %. It should be noted that there are differences among the observations.

(20)P12, L249: Why significant? What is the measure for rating a change significant or not significant?

Reply(20): We agree the use of this word here is confusing. We have now reworded this at L210-211 as: "The RF estimates show that the aerosols have produced cooling at the TOA and surface over the Arabian Sea".

(21)P15, L320: To my opinion it is misleading to talk about convection. I think from the CNDC and ICNC distribution you see where you have clouds.

Reply(21): Thank you for pointing this out. As suggested, we have now removed the discussion on CNDC and ICNC and we repeated and expanded the analysis on isentropic levels. As mentioned in the reply (2) there are two pathways for South Asian aerosol.

(22)P15, L327-328: This is not clear at all. How do aerosols induce a circulation? This relationship needs to be better elaborated.

Reply (22): As per the new analysis on isentropic levels above discussion is obsolete and hence removed.

(23) P17, L367: Compared to heating rates in the tropics that reach several K d-1 these changes are quite low. I cannot follow why this should be a severe or significant change in heating rates.

Reply (23): Heating rates are now expressed in K month$^{-1}$ and percentages to indicate their impact.

(24) P18, L392: As stated above. This is really difficult to believe since the numbers are so low. It would be worth to give the changes additionally in percent so that is easier to follow what this means.

Reply(24): Heating rates are now expressed in K month$^{-1}$ and percentages to indicate their impact.

(25) P18, L412: Same holds for water vapour. Please add also changes in percent.

Reply (25): As suggested, water vapour is now expressed in percentage change.

(26) P20, L444 and Figure 8: I do not see a connection between aerosols in the LS and water vapour entry into the LS. The entry usually appears where you have overshooting convection. Why have you then such a huge amount of aerosols in the LS, but much less water vapour entering in the LS? Have you taken into account that there are also natural processes for aerosols? Especially sulfate aersols are formed naturally in the UTLS (Brock et al., 1995). What kind of aerosols are left if you switch of all aerosols? The natural ones? Unfortunately, it is not clear what exactly is shown in this figure since it is nowhere clearly stated what you derive when you switch of certain aerosols in your model experiments.

Reply (26): See reply(2). In this study, we have perturbed South Asian anthropogenic aerosols only. The major changes are due to anthropogenic aerosols and partly by secondary aerosols which are produced by changes induced by changes of the anthropogenic aerosols It is described L148-155 and in Tabe-1.

(27) P20, L445: References should be added here.

Reply(27): The above lines are deleted in the revised manuscript hence we could not add a reference.

(30) P21, L464: A correlation of 0.57 is not that good. It seems the correlation is generally higher for the Antarctic than the Arctic. Why?

Reply(30): Our analysis on isentropic levels show transport to the Arctic only. The new analysis shows correlation values varying between 0.55 to 0.85.

(31)P21, L464: Only because there is a correlation it does not mean that there is necessarily a connection.

Reply(31): In our model simulations, aerosols are switched on and off over South Asia. Hence variation of aerosol/heating rates at the Artic is a response of South Asian aerosols. Therefore the correlation between aerosol and heating rates indicates a connection.

(32) P21, L466: transported → increase in aerosols?

Reply(32): It is corrected at L436.

(33) P21, L467: It not good to cite only the own papers. Here, definitely also some independent studies should be cited.

Reply (33): We have now added the below mentioned references (See L413-414).

Zheng, C., Wu, Y., Ting, M., Orbe, C., Wang, X., and Tilmes, S.: Summertime transport pathways from different northern hemisphere regions into the Arctic. Journal of Geophysical Research: Atmospheres, 126, e2020JD033811. https://doi.org/10.1029/2020JD033811, 2021

Shindell, D. T., Chin, M., Dentener, F., Doherty, R. M., Faluvegi, G., Fiore, A. M., Hess, P., Koch, D. M., MacKenzie, I. A., Sanderson, M. G., Schultz, M. G., Schulz, M., Stevenson, D. S., Teich, H., Textor, C., Wild, O., Bergmann, D. J., Bey, I., Bian, H., Cuvelier, C., Duncan, B. N., Folberth, G., Horowitz, L. W., Jonson, J., Kaminski, J. W., Marmer, E., Park, R., Pringle, K. J., Schroeder, S., Szopa, S., Takemura, T., Zeng, G., Keating, T. J., and Zuber, A.: A multi-model assessment of pollution transport to the Arctic, Atmos. Chem. Phys., 8, 5353–5372, https://doi.org/10.5194/acp-8-5353-2008, 2008.

(34) P22, L489 and 491: References should be given for these statements. Why do South Asian aerosols enhance water vapour globally?

Reply(34): As per the new analysis on isentropic levels above discussion is obsolete and hence removed.

(35) P23, L518-519: For me these increases in heating seem to be not that strong. Is this really a severe change? From where do you got these numbers? From your study or from the literature. A 2% change in cloud cover anomalies seems to be negligible for me, however, the 12% are rather non-negligible. Here, it also needs to be elaborated more when the changes are significant and when not.

Reply (35): Heating rates are now expressed in K month$^{-1}$ and percentages. The values of heating rates vary between 4 - 60 % which indicates their impact. We have also shown 95% significance level to show changes are significant.

(36) P23, L524: What is the process behind that? Give also here a short explanation.

Reply(36): The above stated text is removed in the revised manuscript.

(37) P23, L527: How do aerosols increase evaporation? Can they also have an effect on condensation? If they affect heating I would rather expect a connection to condensation than evaporation.

Reply(37): As per the new analysis on isentropic levels above discussion is obsolete and hence removed.

(38) P23, L531: Until the end, it is not really explained why BC or other aerosol increase water vapour.

Reply(38): The heating of the troposphere by the carbonaceous aerosol (mainly BC) increases temperature and thereby tropospheric water vapor amounts over the North Indian Ocean and adjoining regions. It is mentioned at L487-489.

**Technical corrections:**

P3, L63 and throughout the manuscript: south Asia or South Asia? You should choose one way of writing and use this consequently throughout the manuscript.

Reply: It is corrected throughout the manuscript.

P4, L76: compared to rest of the Indian region -> compared to "the" rest of the Indian region

Reply: It is corrected, now at L76.

P5, Fig. 1 caption: of year 2016 -> for the year 2016

Reply: It is corrected, now at L99.

P5, L105: Based -> based

Reply: It corrected, now at L106.

P5, L110: delete "the" so that it reads "by convection"

Reply: It corrected, now at L111.

P6, L119: Change sentences as follows: "We use the state of the art aerosol-chemistry-climate model ECHAM6-HAMMOZ."

Reply: It corrected, now at L125.

P6, L127: Instead of just ice-nucleating particles, I would suggest to write (to be more precise) "as kernel for ice-nucleating particles"

Reply: It corrected, now at L135.

P7, L134: replace "the" by "a" → the model simulations are performed with a T63 spectral resolution

Reply: It corrected, now at L142.

P7, L137: at a time step → with a time step

Reply: It corrected, now at L145.

P8, L162: add "are used" at the end of the sentence.

Reply: It corrected, now at Section S1, L19.

P8, L163: add "a" twice → …….measures a radiance…….at a spectral resolution……

Reply: It corrected, now at Section S1 L20.

P9, L173: add "a" and "of" → at a spatial resolution of 0.5° x 0.5°

Reply: It corrected, now at Section S1 L30-31.

P9, L178: add "the" → We evaluate the model performance…….

Reply: It corrected, now at Section S2 L36.

P9, L188: rephrase sentence as follows: The differences are due to uncertainties in the model transport processes, the emission inventory, and the parameterizations.

Reply: It corrected, now at Section S2 L42-44.

P9, L190: add "the" → uncertainties in the satellite measurements

Reply: It corrected, now at Section S2 L45.

P11, Figure 3: Figure size should be increased, so that the scale is better readable.

Reply: We improved the size of figure 3, now scale is better readable.

P11, Figure 3 caption: during → "for the years" or "for the time period from"…..

Reply: It corrected, now at L260.

P11, L231: in → of

Reply: It corrected, now at L195.

P12, L246: "and at the surface" or "and the surface"

Reply: It corrected, now at L208.

P12, L249: add "the" and "a" → show that the aerosols have produced a significant cooling

Reply: It corrected, now at L211.

P12, L254: add "an" → an atmospheric warming

Reply: It corrected, now at L216.

P12, L261: add "the" twice → at the TOA (…) and the surface

Reply: It corrected, now at L224.

P13, L267: add "a" → lead to a heating

Reply: It corrected, now at L228.

P13, L269: Analyses of the → The analyses of the

Reply: It corrected, now at L230.

P14, Figure 4 caption: replace twice "during" by "for the years" or "for the time period"

Reply: Now it is Figure 3 and it corrected at L260.

P15, L318: add "from" → from north India

Reply: As per the new analysis on isentropic levels above sentence is removed.

P18, L409: add "the" → on the water vapor distribution

Reply: It corrected, now at L391.

P18, L417: abortion → absorption

Reply: As per the new analysis on isentropic levels above sentence is removed.

P19, L432: add "the" → over "the" Arabian Sea

Reply: It corrected, now at L403.

P19, L432: add "over" → averaged over

Reply: It corrected, now at L406.

P19, L433: during → "for the years" or "for the time period"

Reply: It corrected, now at L407-408.

P20, L448: it → This figure

Reply: It corrected, now at L420.

P20, L449: add "the" → during spring and the monsoon seasons

Reply: It corrected, now at L421.

P21, Figure 21: add "in the" → "in the UTLS"

Reply: It corrected, now at L446.

Replies to Reviewer-II

Fadnavis et al. discuss the broader climatic impacts of eliminating South Asian emissions, examing the atmospheric response from pole-to-pole and upward into the stratosphere. The results stimulate a number of interesting questions and I enjoyed reading the paper, but still there are some logical and contextual gaps that need to be addressed, especially in section 3. These are perhaps more important because the results are based on integration of a single climate model under different emissions scenarios. The simulations are very much worth analyzing and reporting, but the presentation leaves me sceptical of some of the conclusions for this model specifically along with the generalizability of the results to the natural atmosphere. Addressing questions of generalizability is beyond the scope of this paper (though I appreciate the discussion along these lines in section 3.1), but the reliability of the conclusions for this model system can be addressed more comprehensively.

 In addition to my suggestions for this paper below, I have tried to outline some ideas and questions that inspired me while reading the paper. I set out to be brief but did not always succeed. The authors should take these suggestions not as "I want you to include all of these in this paper" but rather "you should make this paper more coherent, and may want to consider these ideas for future work". If the authors want to discuss any of these ideas further, they are very welcome to contact me! Please see also optional editorial suggestions in the annotated draft.

Reply: We thank the reviewer for their careful assessment of our study, the valuable suggestions, positive comments, and for sharing ideas for future work. We have incorporated all suggestions into the revised manuscript. We have performed an analysis on isentropic levels. Accordingly, we have modified section 3, abstract, and conclusions. We have avoided generalizability in the revised manuscript.

The changes are indicated in blue colour in the manuscript at the line numbers mentioned in the replies.

**General comment 1:** The concept of the aerosol-induced secondary circulation introduced in the paper is an intriguing one, but the description of this is unclear and potentially misleading. If I were to only read the abstract or conclusions, I would imagine a large anomalous overturning that links the convective regions over the Bay of Bengal and the Arabian Sea all the way to the Southern Hemisphere. However, this picture does not match either the circulation response as illustrated in Figure 5 or the background circulation during March-May. In particular, Figure 5 does not support the statement "In the UTLS, [the aerosol] are further transported to the southern hemisphere and downward into the troposphere." I can see a little bit of what you are talking about in the supplementary figure, but I am skeptical of this description and it needs to be better justified. I'll outline an alternative hypothesis below, both to illustrate why more justification is needed and as an idea that might be interesting for you to explore further.

Reply(1): Thank you for this point as we agree it is important to explain the overall concept more clearly. We have now performed an additional analysis on isentropic levels and we provided a justification for the transport of aerosols into the Southern hemisphere as below (see L25-28, Section 3.3, and L286-303)

Our analysis indicates that the Hadley circulation (Fig. 5a and Fig. S3) with its ascending branch over the Indian Ocean and adjoining region (60º E – 140º E, 0 – 30º N), lifts the South Asian aerosols to the UTLS. These aerosols enter the westerly jet (Fig. 4 d-f). The distribution of zonal winds in Fig. 5b shows transport into the southern hemisphere preferentially in regions of equatorial westerly winds, so-called "westerly duct" regions (Waugh and Polvani, 2000; Yan et al., 2021), where Rossby-wave breaking occurs (Fig. 5b and Fig. S4). This is consistent with findings from Frederiksen et al. (2018) who have also shown interhemispheric transport of $CO_2$ via Pacific and Atlantic westerly ducts during the spring season. Fig. 5c shows that changes in South Asian aerosols concentrations cause a shift in the Pacific duct. Thus interhemispheric transport occurs through (1) an Atlantic duct and (2) a slightly shifted Pacific duct (5º S – 5º N, 50º E – 140º E), i.e. over the Indian-Ocean-Western Pacific region (also see Fig. 4 d-f). The shift in Pacific duct in a response to South Asian aerosol changes is likely due to higher Rossby wave bearing near south Asia. The geopotential (Fig 5d) and potential vorticity (Fig. S5) anomalies (CTL-Aerooff) show Rossby wave breaking near the Indian-Ocean-Western Pacific region that could lead to Southern hemispheric transport through the Indian-Ocean-Western Pacific region path (Fig 5 d-e).

(2) The ITCZ in the Indian Ocean region is still located in the Southern Hemisphere in March and early April, migrates north to near the equator from late April to early May, and then slowly proceeds north into the Asian summer monsoon region over late May and most of June (see, e.g., Figure 2 of Schneider et al 2014). In this context, the tropical circulation responses look more like a weakening of the tropical overturning Hadley-type circulation in this region. How could the aerosols produce this response? My guess: the aerosol enhancements in the tropics are mostly located near the surface. The cross-equatorial flow that feeds the ITCZ is anticyclonic in the hemisphere containing the ITCZ, and therefore must be cyclonic in the upstream hemisphere. The effect of the relatively shallow aerosol layer on radiative heating will represent an anticyclonic vorticity source upstream, which should tend to weaken moisture supply from the Northern Hemisphere, and might also delay the northward propagation of the ITCZ in May by making the environment just north of the equator less favorable for the ITCZ

to move into. Hoskins et al. (2020) and Hoskins and Yang (2021) provide very clear explanations of these processes for the solstice-season ITCZs that might be useful.

Aerosol changes in the subtropical Southern Hemisphere might also have an impact, maybe by disrupting the effects of extratropical waves on intraseasonal active and break phases along the March-April ITCZ (much of the tropical MJO appears to be driven by moisture advection modulated by SH wave activity; see e.g. Li 2014). This hypothesis would be consistent with the opposing vertical velocity responses in the 60-75E and 75E-90E bands around 40S. We cannot tell the characteristics of that anomaly in the extratropical wave, but you could check it in maps of upper level geopotential height and winds to see if it matches the hypothesis articulated by Li (2014). From what I've read I think this explanation for the Indian Ocean ITCZ change is more likely than the first, but both might play some role.

Reply (2): We thank the reviewer for the thought-provoking ideas, discussions, and important references. We have analysed the monthly variation of the vertical velocity field (Fig 5a, Fig. S3). It shows ascending winds over the North Indian Ocean – Western Pacific (65º E – 140º E) lifting the South Asian aerosols to the UTLS during the months from March to May. These aerosols enter the westerly jet in the northern hemisphere. They are further transported to the Southern Hemisphere and downward (320 – 340K) via an equatorial Atlantic westerly duct (5º S – 5º N, 10º W – 40º W) and shifted westerly Pacific duct (5º S – 5º N, 95º E – 140º E). The shifting of a Pacific westerly duct may be due to higher Rossby wave breaking caused by the South Asian aerosol. It is discussed in section 3.3.

(3) In addition to the lat-long distribution of upper-level response in the southern hemisphere, there are two relatively straightforward things that you could do to check these possibilities. First, compare the circulation response month by month to the seasonal mean response. Maybe even step through the full seasonal cycle with reference to the changing location of the ITCZ, since the solstitial dynamical responses might be easier to interpret and can then be linked to the springtime transition. Second, you might look at the evolution of the circulation response over time. If I have understood correctly you start all simulations from the same inital dynamical conditions, with one year of spinup to introduce and equilibrate to the emissions perturbations. My guess is that the aerosol changes in the southern hemisphere should be accumulating over time, and would be mainly linked to pulses of supply via that boreal wintertime cross-equatorial flow into the ITCZ. If this is indeed the case, then you should see a strong adjustment in the circulation over the first couple of years, including the spinup year. If either of these helps to explain the changes, then it might be worth including them in either the main text or supplement.

Reply(3): Thank you for this suggestion. We have now included an explanation of the mechanism for the transport of South Asian aerosols to the southern hemisphere in reply (2) and in section 3.3 at L289-303.

(4) The question is then: how to explain the protrusion of increased aerosol (especially BC and OC) in the tropical upper troposphere, which then seems to get sucked toward lower levels in the southern hemisphere tropics. Here I think it is again helpful to remember that we are looking at circulation anomalies and that the ITCZ may be weaker but there is still a lot of convection there. Another thought is that, if you look at the mass streamfunction for the overturning circulation in the tropical southern hemisphere, there are really two overturnings, one linked to cumulus congestus that diverges around 400-500 hPa and one linked to deep convection that diverges around 200-300 hPa. These patterns in the aerosol are rather reminiscent of that, and

so I wonder if it just indicates entrainment into that spectrum of convection a little bit above the glaciation level, which then invigorates the convection in the middle-to-upper troposphere and enhances aerosol wet deposition (the negative anomalies above and below). Here some details of the model are important: is aging of BC and OC represented, increasing the hydrophilic fraction? Are mixed phase clouds permitted, and how is the partitioning of liquid and ice represented in these? What are the roles or efficiencies of BC and OC as CDNC and IDNC?

This is all just speculation, and it's kind of strange to think about an ITCZ that is somehow both weaker/stabilized in the lower troposphere and more intense/destabilized in the upper troposphere. However, the main point is just that you need more justification and explanation to support your description of the 'secondary circulation' response.

Reply (4): Since analysis is performed on isentropic levels the protrusion of increased aerosol in the tropical upper troposphere, which is then transported toward lower levels in the southern hemisphere, is visible in Figure 5. Our analysis shows that southward transport is associated with Rossby wave breaking in the westerly jet causing the transport of South Asian aerosols to the southern hemisphere via the Atlantic westerly duct and shifted Pacific westerly duct as stated in reply (2). This is discussed in section 3.3.

**(5) General comment 2:** It is difficult to keep track of all the comparisons in section 3.2. Part of this is the presentation, which is very full of numbers, and part of this is the lack of detail about differences in measurements or methologies across the studies being referenced. For example, the three studies mentioned in L259-263 report in-atmosphere aerosol forcing that is an order of magnitude larger than yours. Is this just the difference in winter versus spring, e.g. in aerosol loading or solar zentith angle or both? Was there particularly strong burning during the years they measured that enhanced the relative loading of black carbon? Were they just overestimating the fraction of BC in the column? It would benefit the paper a lot to include more context and comparison here beyond just the quantitative results.

Reply (5): Thank you for the suggestion. We have removed some of the references. There are currently only sparse observations over the Indian region. Here we want to state that past studies show negative RF at the TOA and surface. While in-atmospheric RF is positive. To explain differences we have added "There is a large variation in the magnitude of RF (at the TOA, surface, and in-atmosphere) reported from observations and our model simulations. This may be due to different regions and different time periods and the relatively coarse model resolution. L224-226.

**(6) General comment 3:** Section 3.4 is missing some important context. For example, I was well into the section before I realized that the heating rates are total heating rates, rather than radiative heating rates. First, I think that some background context would be helpful for the heating rates. How large are these differences in heating rates relative to values in Aerooff or CTL? Where are the changes opposing the mean heating as opposed to acting in the same direction? Are they tendencies in temperature or potential temperature? If temperature, it might be worth considering a switch, since much of the focus is on changes in the UTLS. It would also be great to have more decomposition of the heating rates (i.e., SW + LW + non-rad or SWclr + LWclr + clouds and turbulence).

Second, I think background context would also be helpful for the water vapor changes. Since the spatial gradients in water vapor volume mixing ratio are large, changes could be reported

in % relative to Aerooff or CTL. This might also be more physically meaningful given the logarithmic dependence of water vapor's greenhouse impact as a function of concentration. Third, many of the interpretations are difficult to judge as a function of simply these Eulerian cross-sections in longitude-pressure. The similarities between the Arabian Sea cross-section and the zonal mean could be taken to mean that 'this slice dominates the response' or they could be taken to mean 'zonal advection is efficient'. In the latter case, how much can we trust some of the meridional features you are highlight, such as the pathway of enhanced aerosol extending southward and toward the surface and its possible effects on water vapor? Finally, transit times to several of the regions you highlight in the stratosphere, both at low and high latitudes are several months at least. You do mention this at one point (L468-469), but I'd recommend to mention it earlier and more often because this is essential context that some readers may not be familiar with. In any case, these long transit times cannot be related to cross-tropopause exchange during spring on the evidence presented in this paper, and indeed the seasonal cycles shown in figure 8 suggest that cross-tropopause transport in this region is pretty weak during March and April. Given these limitations, why focus so much on the stratospheric response in a paper that is rooted in changes over the North Indian Ocean during MAM?

Reply (6): As suggested we have decomposed the heating rates (i.e., SW and LW). The model does not provide heating rates for SWclr + LWclr + clouds and turbulence etc. The simulated heating rates show that short wave heating due to carbonaceous aerosols is the major reason for heating in the path of transport of aerosols. Black carbon aerosol produces higher heating than organic carbon aerosols (see Figure below). In the manuscript, we have included net heating rates (SW+LW) to limit the number of figures. While we have mentioned "Black carbon aerosol produces higher heating than organic carbon aerosols. The shortwave heating due to BC aerosols is the major contributor to the total heating (Fig. not included)." L368-370.

In the revised analysis, there are two pathways for inter-hemispheric transport, an Atlantic duct, and a duct over the Indian Ocean-western Pacific region. Hence, we have shown a cross-section plot over the region coving two branches, i.e. $30^{\circ}$ E – $140^{\circ}$ E. The new analysis on the isentropic level does not show transport from UTLS to the surface in the Southern hemisphere. Hence discussion on heating by aerosols and its effect on water vapor in the Southern hemisphere is removed in the revised version.

The concentration of changes in the water vapor volume mixing ratio is now expressed in percentages. The discussion on these long transit times is now shifted earlier at L417-418.

Since we have revised the analysis on isentropic levels, the new results show a cross-tropopause transport of aerosols and water vapor during spring (March-May) and the monsoon seasons (Fig. 8).

The analysis of the isentropic level shows the transport of South Asian aerosols in the UTLS hence discussion on the deep stratosphere is removed. We have also mentioned that "It should be noted that increase in aerosols to the Arctic also occurs during the monsoon season (Fadnavis et al., 2017a, 2017b, 2019, Zheng et al., 2021) which may affect the dynamics and aerosol amounts in the spring of the next year in the UTLS." L437-439.

**Total heating rates (LW+SW)**

[Figure]

Figure: Zonal section (averaged $30^{\circ}$ E – $140^{\circ}$ E and for spring) of net heating rate (LW+SW), (Kmonth$^{-1}$) from simulations (a) CTL-Aerooff, (b) CTL-noBC, (c)CTL-noOC, (d) CTL- nosul, (e-h) same as (a-d) but for SW heating rates, (i-l) same as (a-d) but for LW heating rates.

**(7) General comment 4:** Many of the figure elements are too small for me to discern. It should be possible to improve most of these by modifying axis ranges (e.g. zooming in on the regions that are highlighted in the text), axis styles (e.g. linear to logarithmic or vice versa), and internal elements of the plot (e.g. vectors versus streamlines, density of vectors, etc.).

Reply(7): As suggested the figures in the paper are now improved.

**(8) References:**

Hoskins, B. J. & Yang, G.-Y. The Detailed Dynamics of the Hadley Cell. Part II: December–February. J Climate 34, 805–823 (2021).
Hoskins, B. J., Yang, G. â⬛⬛Y. & Fonseca, R. M. The detailed dynamics of the June–August Hadley Cell. Q J Roy Meteor Soc 146, 557–575 (2020).
Li, T. Recent advance in understanding the dynamics of the Madden-Julian oscillation. J Meteorol Res 28, 1–33 (2014).
Schneider, T., Bischoff, T. & Haug, G. H. Migrations and dynamics of the intertropical convergence zone. Nature 513, 45–53 (2014).

Reply(8): Thank you for suggesting the above references.

**Specific comments:**

(9) I suggest to specify March-May after "spring" in the abstract.

Reply (9): Above suggestion is incorporated in the revised manuscript at L22

(10)There are a lot of regions to keep track of in the paper; it might be good to eliminate all use of "the North Indian Ocean" and always use "the Arabian Sea", "the Bay of Bengal", or "the Arabian Sea and Bay of Bengal"

Replay (10): In the revised manuscript, we have averaged aerosols over area $30^\circ$ E – $140^\circ$ E (includes the Arabian Sea and the Bay of Bengal). Hence we have used the term the North Indian Ocean in sections 3.3 and 3.4.

(11) General note for region descriptions: South India or North Indian Ocean should use capitalized South and North, but southern India and northern Indian Ocean would generally not.

Reply (11): The above suggestion is incorporated in the revised manuscript.

(12) L55-57: I am not sure I understand what these numbers mean -- 97% of what? It means that only 3% were in the coarse mode? It would be helpful to clarify.

Reply (12): The above sentence is revised as "Several other in situ observations, e.g. over the Maldives during November 2014 – March 2015, show that air masses arising from the Indo-Gangetic Plain contain very high amounts (97 %) of the elemental carbon in the $PM_{10}$ was found in the fine mode" L54-56

- L72: Check number formatting here, I think this should be x10^n

Reply: It is corrected now at L72-73.

- L121: Might be useful to write out these abbreviations (e.g. "HAM") for those interested in the model

Reply: It is now mentioned Hamburg (HAM) at L127.

- L122-123: Here is another place where simplification might help -- is there a reason to use POM here and OC elsewhere in the manuscript?

Reply: It is modified now as "organic carbon (OC)" at L129.

- L127: According to prescribed microphysical properties or all aerosol are treated equally?

Reply: The above sentence is modified as "HAM explicitly simulates the impact of aerosol species on cloud droplet and ice crystal formation according to prescribed microphysical properties." L132-L135.

- L167: There should be a citation for MODIS Terra AOD; probably it is this one: MODIS Atmosphere Science Team: MODIS/Terra Aerosol Cloud Water Vapor Ozone Monthly L3 Global 1Deg CMG, https://doi.org/10.5067/MODIS/MOD08_M3.061, 2017\. This doi citation should be used in place of the link because it is fixed to the dataset and version.

Reply: The above suggestion is incorporated at Section S1 L25.

- L170: Here too, probably: Diner, David: MISR Level 3 Global Joint Aerosol monthly product V002, https://doi.org/10.5067/TERRA/MISR/MIL3MJTA.002, 2020.

Reply: The above suggestion is included at Section S1 L32.

- L178: Are simulated aerosol processes processed to support like-for-like comparison with the model (e.g. cloud clearing)? Could this matter for validation?

Reply: The above mentioned discussion is moved to supplement as suggested by the reviewer-I. The model output is not cloud-free. There are uncertainties in the model processes and satellite measurements (it is mentioned in the supplement at Section S2, L36-46). Here, we wish to show that model could simulate overall features.

- L190: Can all of the potential sources of bias between CTL and the observed AOD be expected to scale in simple and consistent ways across the sensitivity simulations?

Reply: Most of the biases are the same in CTL and sensitivity simulations but not all. Hence we have mentioned, "With model biases present in both the CTL and the perturbed simulations, investigating anomalies removes **some of the model bias**." (section S2, L46-48).

- L194: "Fair" is a rather equivocal word to use here; I read it as you think the model performance is not particularly good, but there are a number of other ways to read it and this may not be what you mean at all. It would be helpful to be more specific here.

Reply: It is now removed in the revised manuscript and supplement.

- Fig2: These spatial distributions are more different than what I had expected from the text, but I think that is because you are only evaluating the AOD over South Asia and surrounding seas. Maybe zoom in to 45E-100E and equator to 40N?

Reply: Here we want to show the transport of South Asia aerosols to the Western Pacific and towards the Equator. Hence Figure 2 is limited to 10° S - 40° N, 55° E - 150° E.

- L204: Should be section 3

Reply: It is modified now.

- L206: This sentence should be rephrased for clarity, the distributions are not from Aerooff, they are from the difference between CTL and Aerooff, right?

Reply: It is corrected at L171.

- L260: Please check the dates, it looks like both studies here included at least a couple of years in addition to 1999

Reply: We have removed the study by Satheesh and Ramanathan 2000; Rajeev and Ramanathan et al, 2001.

- L279: Here the regions meant are again a little vague, especially in comparison to some other parts of the paper

Reply: To limit the region Figure3 is now plotted over the Indian region.

- L320-324: It feels like Fig S2 is doing a lot of work here; why put it in the supplement instead of the main text?

Reply: We have now added Figure 5a indicating ascending winds over the Indian region similar to the old Fig S2.

- Fig5: The vectors and other key elements here are very small. I understand why it is included in later figures, but is it necessary to include the 10-50 hPa part of this figure? This is not really discussed and the differences are small. It would be helpful to eliminate this and make the aspect ratio larger in the horizontal direction as well.

Reply: Since analysis is shown on isentropic levels hence the old Fig 5 is removed.

- L393: This makes sense if the additional heating is from additional latent heating above the glaciation level, but how do you attribute it to that process specifically?

Reply: Since analysis is shown on isentropic levels the above point is obsolete.

- L395: Is this unbroken channel of ascent entirely convective, or mainly convective with radiative heating balancing adiabatic cooling in slow ascent above?

Reply: Since analysis is shown on isentropic levels the above point is obsolete.

- L398-399: It seems plausible that this water vapor is transported from the Arabian Sea, but it's very difficult to judge this conclusion from Eulerian slices/zonal means alone. Also, I'm not sure if I'm missing something, but I cannot readily make this connection through comparison of fig 7 to fig 5. Please clarify the logic behind interpreting the results this way.

Reply: Since analysis is shown on isentropic levels the above point is obsolete.

- L403: Since this sentence refers to the impact of temperature change on water vapor, how do you link this effect specifically to the heating caused by carbonaceous aerosols? How do you rule out adiabatic warming of air containing aerosol, or mixing near the tops of slightly invigorated cumulus congestus below?

Reply: Since analysis is shown on isentropic levels the above point is obsolete.

- L406: Please specify how 'lower stratosphere' is defined in this sentence, in terms of the vertical coordinate.

Reply: Lower stratosphere is defined as 380K-400K. (L351)

- L407: It seems possible that the water vapor increases in the high-latitude stratosphere could be more about aerosol effects in mid-latitudes and how they modify wave activity propagating upward in the springtime polar vortex; given the season, maybe even in the timing of final warming (in the NH) or vortex strengthening (in the SH).

Reply: Since analysis is shown on isentropic levels the above point is obsolete.

- L417: typo

Reply: Since analysis is shown on isentropic levels the above sentence is changed.

- L417: Is the sulfate impact on longwave radiation mentioned here explicitly diagnosed? What determines this impact, and how can a 'negligible impact' lead to water vapor **enhancement through such a deep layer?**

Reply: we have mentioned that "The water vapor enhancement by sulfate aerosols ~0.2 - 1% in pockets (Fig. 7d)." L393-395.

- Fig 7: Please make the distinction between (b), (d), and (f) clearer. I spent way too much time puzzling over how the zonal mean response (f) in the high-latitude stratosphere could be so different from the 55-70E section (e)!

Reply: Since analysis is shown in isentropic levels the above figure is modified.

- L465-466: Is diabatic heating in the polar lower stratosphere positive or negative in the equinoctial seasons?

Reply: Since analysis is shown in isentropic levels the above point is obsolete.

- L490: This should be true for LW heating in the troposphere below the level where optical depth to TOA ~ 1, but the LW effect of increasing concentrations above that point should be negative (enhanced emission across the water vapor bands with less coming back than is emitted). More water vapor through the whole troposphere should shift the optical depth ~ 1 level upward, so does deepen the layer where water vapor increases positively impact heating, but the upper troposphere should remain above it. Enhanced heating through the upper troposphere may thus be linked more to increased SW absorption or increased latent heating. If you have the individual terms from your model, it may be worth looking at them.

Reply: Thank you for the suggestion. We plan to elaborate on this in the new study.

- L494: specify whether this forcing is at surface or at TOA

Reply It is at the TOA. It is now mentioned at L457.

- L501: should specify South Asia here, right?

Reply: The above suggestion is included at L465.

- L505: "0" should be "o".

Reply: The above suggestion is included at L468.

- L515-516: maybe put the latter part of this sentence (+4.33 ...) in parentheses and then add "alone" to avoid confusion with the Aerooff results discussed at the beginning of the paragraph

Reply: The above suggestion is included at L485.

- L516-519: Which troposphere is referenced here, Arabian Sea or Indo-Gangetic plain? Both? I'm still not sure why the tropospheric heating acts to intensify convection rather than stabilize the atmosphere; shouldn't this be considered more as a result of convective invigoration rather than a cause? From Fig S2 it does look like there is some intensification of deep convection above the glaciation level over the western Arabian Sea and Arabian Peninsula, but the opposite seems true over the Bay of Bengal. Then again, I am having some trouble reconciling figures S2 and S3, so maybe I have misunderstood?

Reply: Since analysis is shown in isentropic levels the above point is obsolete.

- L527-528: Is it increased evaporation or increased temperature that leads to this? How does the balance of P-E change?

Reply: Yes, due to increased temperature. It is now corrected at L488-490.

**Supplement:**

- L58: There is an extra space between m and g

Reply: Now it is expressed in %.

- Fig S2: what are the vectors? I have assumed that they are the vertical and zonal motion, which seems consistent with OLR, but not with figure S3.

 Reply: Since analysis is shown on isentropic levels the above point is obsolete.

- Fig S3: How to reconcile stronger ascent over 10-20N in the Bay of Bengal section with reduced cloud cover, enhanced OLR, and CDNC+ICNC as shown in figure S2?

Reply: Since analysis is shown on isentropic levels the above point is obsolete.

- L101: "anomalies aerosols" -> "aerosol anomalies"?
Reply: Since analysis is shown on isentropic levels the above point is obsolete.